

# Maternal ancestry analyses of red tilapia strains based on D-loop sequences of seven tilapia populations

Bingjie Jiang[1], Jianjun Fu[2], Zaijie Dong[1,2], Min Fang[1], Wenbin Zhu[2] and Lanmei Wang[2]

[1] Wuxi Fisheries College, Nanjing Agricultural University, Wuxi, Jiangsu, China
[2] Key Laboratory of Freshwater Fisheries and Germplasm Resources Utilization, Ministry of Agriculture and Rural Affairs, Freshwater Fisheries Research Center of Chinese Academy of Fishery Sciences, Wuxi, Jiangsu, China

Corresponding author
Zaijie Dong, dongzj@ffrc.cn

## ABSTRACT

**Background.** Many tilapia species or varieties have been widely introduced and have become an economically important food fish in China. Information on the genetic backgrounds of these populations is deficient and requires more research, especially for red tilapia strains.

**Methods.** In the present study, displacement loop (D-loop) sequences were used to evaluate the genetic relationship and diversity of seven tilapia populations that are widely cultured in China; this was done specifically to speculate on the maternal ancestry of red tilapia strains. Three red tilapia varieties of *Oreochromis ssp.*, Taiwan (TW), Israel (IL), and Malaysia (MY) strains and other populations, including *O. aureus* (AR), *O. niloticus* (NL), *O. mossambicus* (MS), and the GIFT strain of *O. niloticus*, were collected and analyzed in this study.

**Results.** A total of 146 polymorphic sites and 32 haplotypes of D-loop sequences were detected among 332 fish and four major haplotypes were shared among the populations. The TW and NL populations had a greater number of haplotypes (20 and 8, respectively). The haplotype diversity (Hd) and nucleotide diversity ($\pi$) of each population ranged from 0.234 to 0.826, and 0 to 0.060, respectively. The significant positive Tajima's D value of neutral test were detected in the NL, IL, and MY populations ($P < 0.05$), which indicated these populations might have not experienced historical expansion. According to the pairwise $F$-statistics, highly significant genetic differentiations were detected among populations ($P < 0.01$), with the exception of the IL and MY populations ($P > 0.05$). The nearest K2P genetic distance ($D = 0.014$) was detected between the MS and TW populations, whereas, the farthest ($D = 0.101$) was found between the GIFT and AR populations. The results from the molecular variance analysis (AMOVA) showed that there was an extremely significant genetic variation observed among the populations ($P < 0.01$), which contained 63.57% of the total variation. In view of the genetic relationship of red tilapia strains with other populations, TW and IL were detected with more similar genetic structures related to MS, and MY was more genetically similar to GIFT (or NL), which could provide more genetic evidence for the red tilapia strains maternal ancestry.

## INTRODUCTION

Tilapia as a common name has been applied to various cichlids from three distinct genera, which include *Oreochromis*, *Sarotherodon,* and *Tilapia* (*Trewavas, 1983*). The farmed tilapia production worldwide was over 5.8 million tons annually in 2017 and China is the largest tilapia producer in the world (*FAO, 2019*). Red tilapia is a name used for several different man-made tilapia variants that have an attractive red coloration. These variants are the result of continuous selective breeding (*Wohlfarth et al., 1990*). Many farmers prefer to cultivate red tilapia since it is sought after in certain markets. Because of their high protein content, large size, high feed conversion rate (FCR), rapid growth and palatability, red tilapia is the focus of major farming efforts in China (*Romana-eguia & Eguia, 1999*).

In recent years, due to the increasing demand in the market, many red tilapia populations have been imported and cultured in Chinese farms. However, genetic introgression of those varieties is commonly detected because of their inter-population hybridization breeding. Growth and color separation often occurs in this practice, which greatly affects the promotion and marketing of red tilapia. In China, the genetic diversity studies of the tilapia populations based on molecular markers were carried out in the tilapia populations using TRAP (*Ma, 2012*), microsatellites (*Zhang et al., 2010*) and ISSR (*Zhong et al., 2012*). Research on tilapia in other countries has mainly focused on growth and development (*Lith et al., 2005*), culture (*Muendo et al., 2006*), and breeding (*Fujimura & Okada, 2010*). The information on genetic diversity and the genetic ancestries of red tilapia is lacking. The origin of the red tilapia was generally thought to be attributed to the cross-breeding of the mutant reddish-orange *O. mossambicus* and other populations including *O. aureus*, *O. niloticus,* and *O. hornorum* (*Wohlfarth et al., 1990*; *Sandeep et al., 2012*), but the specific source of the certain strains is ambiguous and the ancestries of the three strains widely cultivated in China are uncharted (*Zhu et al., 2017*).

The D-loop sequences are non-coding regions of mitochondrial DNA (mtDNA), with a high rate of evolution and no recombination, which becomes one of the most commonly used mtDNA sequences for addressing the evolutionary relationship of close relatives and/or subspecies (*Murgra et al., 2002*). At present, the D-loop sequences have been widely used in genetic analyses for aquaculture species, especially related to the genetic structure (*Ryota & Akira, 2002*), genetic differentiation (*Brown & Thorgaard, 2002*), species validities (*Tang, Hu & Yang, 2007*), phylogeny and molecular differentiation (*Ekerette et al., 2018*). In this study, D-loop sequences were used to evaluate the genetic diversity of seven tilapia populations and to further estimate the maternal ancestry of three strains of red tilapia.

## MATERIALS AND METHODS

### Sample collection

The sampling scheme and experimental protocols were approved by the Bioethical Committee of Freshwater Fisheries Research Center (FFRC) of the Chinese Academy of Fishery Sciences (CAFS) (BC2013863, 9/2013). Experimental fish were sampled from seven populations, including the three red tilapia strains of *Oreochromis spp.*, Chinese Taiwan (TW), Israel (IL), and Malaysia (MY) strains, and other populations of tilapia,

including the GIFT strain of *O. niloticus*, *O. aureus* (AR), *O. niloticus* (NL), *O. mossambicus* (MS). The TW and IL populations were transferred from Fujian Province, China in 2014, and the MY population was introduced from Malaysia in 2009 (*Yang et al., 2015*). All tilapia populations were domesticated and bred in an experimental aquaculture farm in Wuxi (Jiangsu Province, China). The methods for handling the animals and the experimental procedures conducted were in accordance with the guidelines for the care and use of animals for scientific purposes set by the Ministry of Science and Technology, Beijing China (No.398, 2006). Forty-eight fin clips were sampled from each population and soaked in absolute ethanol until the DNA was extracted.

## DNA extraction and amplification

Genomic DNA was extracted using the phenol-chloroform method (*Sambrook & Russell, 2001*). The integrity was detected by 1% agarose gel electrophoresis. The purity and concentration of the DNA was detected using the NanoDrop spectrophotometer. The DNA concentration of each sample was adjusted to about 20 ng/$\mu$L and kept under $-20\,°C$ until ready to use.

Primers of the D-loop were designed according to the complete sequences of tilapia mtDNA (Accession NO: NC_014060) from the National Center for Biotechnical Information (NCBI). The D-loop sequences of 867 bp was amplified using a primer pair (sense primer: 5′-CTACTTCTTCCTCTTCCTTGT-3′, anti-sense primer: 5′-TCCGTCTTAACATCTTCAGT-3′), which was synthesized by Sangon Biotech (Shanghai) Co.Ltd. The PCR amplification was performed on an Eppendorf Mastercycler Pro 384 PCR thermocycler (Eppendorf, Hamburg, Germany). Amplifications were performed in a volume of 50 $\mu$L, containing 5 $\mu$L 10× PCR Buffer, 3 $\mu$L MgCl$_2$ (0.25 mM), 4 $\mu$L dNTPs (2.5 mM), 1 $\mu$L Taq polymerase (2.5 U/$\mu$L), 1 $\mu$L of each primer (10 $\mu$M), 2 $\mu$L genomic DNA (20 ng/$\mu$L), and 33 $\mu$L DNase/RNase-free deionized water. PCR amplification was performed under the following conditions: pre-denaturing for 2 min at 94 °C, 35 cycles of denaturing for 40s at 94 °C, annealing for 55s at 55 °C, prolonging 1 min at 72 °C; final prolonging for 10 min at 72 °C; and then held at 12 °C. Subsequently, the reaction products were detected using 1% agarose gel electrophoresis, and the bidirectional sequencing was carried out with the ABI3730XL sequencing instrument of the Shanghai Majorbio Company.

## Sequence arrangement and data analysis

The sequences were edited using the BioEdit version 7.0.9 software (*Hall, 1998*). To ensure accuracy, all DNA fragments were sequenced in two directions, and the assembled sequences were manually checked to prevent the ambiguity of the base or sequencing error. After the completion of the splicing, all sequences were used for homologous alignment and length determination by the BioEdit version 7.0.9 software.

After comparing and sorting the D-loop sequences, the fuzzy sequences were deleted and 332 homologous sequences were obtained upon completion. Genetic variation parameters were calculated by the DnaSP 5.1 software (*Librado & Rozas, 2009*), including polymorphic (segregating) sites (S), number of haplotypes (h), haplotype diversity (Hd),
nucleotide diversity ($\pi$), average number of nucleotide differences (k) and Tajima's D. For phylogenetic analysis, MEGA 5.05 software (*Kimura, 1980*; *Tamura et al., 2011*) was utilized to calculate the Kimura 2-parameter (K2P) distance among populations, and to construct unweighted pair group methods with an arithmetic (UPGMA) dendrogram set with 1,000 replications of bootstrapping. Arlequin 3.5 software (*Excoffier & Lischer, 2010*) was used to analyze the nucleotide composition, $F$-statistics ($F_{ST}$) and the analysis of molecular variance (AMOVA) among seven tilapia populations. The Network 4.6 software (*Polzin & Daneshm, 2003*) was used to construct the network for haplotypes of D-loop sequences.

## RESULTS

### Variation and haplotype distribution of D-loop sequences in tilapia

The nucleotide frequencies of the seven tilapia populations were consistent, and clearly, the rate of A + T (the average value is 64.3%) was higher than in C + G (the average value is 35.7%). By sequence alignment, a total of 32 haplotypes were found in the D-loop sequences and deposited in the GenBank database under the accession numbers MH515150–MH515185 (except for MH515152, MH515172, MH515175, and MH515182). Different numbers of haplotypes (from 1 to 20) were detected among the populations (Table 1). Four of these haplotypes were shared haplotypes (Hap_2, Hap_22, Hap_23, and Hap_24), of which 2 haplotypes were composed of NL and GIFT populations (Hap_22, Hap_23), while the others were unique to each population. The 4 dominant haplotypes in all individuals were Hap_2, Hap_22, Hap_24, and Hap_26 accounting for 34.90%, 12.30%, 14.50%, and 14.50%, respectively.

### Genetic diversity and genetic distance among seven tilapia populations

The genetic diversity parameters of the tilapia populations based on D-loop sequences are shown in Table 2. A total of 146 polymorphism sites were found, the overall haplotype diversity (Hd) of the tilapia populations was 0.817, and each population ranged from 0 to 0.834. The average number of nucleotide differences ($K = 0 - 47.32$) and nucleotide diversity ($\pi = 0 - 0.060$) were determined. Among them, the AR population had the lowest genetic diversity ($Hd = 0$, $\pi = 0$), the TW and NL populations had a higher haplotype diversity ($Hd = 0.834$, 0.826, respectively), and the highest nucleotide diversity ($\pi$) was detected in the NL population ($\pi = 0.060$). Tajima's test indicated that the Tajima's D value of the TW, GIFT, and MS populations were negative, and other populations were positive. Among them, NL and MS reached a significant level ($P < 0.05$), GIFT, IL and MY populations reached extremely significant levels ($P < 0.01$).

The pairwise genetic distance was calculated using the Kimura 2-parameter (K2P) model among seven tilapia populations (Table 3, below diagonal). The inter-population distances among seven populations ranged from 0.014 to 0.101. The closest inter-population distance (0.014) was between the MS and TW populations, while the furthest inter-population distance (0.101) was between the GIFT and AR populations. In this study, the UPGMA tree based on the K2P genetic distance was shown in Fig. 1. According to the phylogenetic tree,

Table 1 Distribution of the D-loop haplotypes in tilapia populations.

| Haplotype | Accession | Wild-type or breeding populations | | | | Red tilapias | | | Sum |
|---|---|---|---|---|---|---|---|---|---|
| | | NL | AR | MS | GIFT | TW | MY | IL | |
| Hap_1 | MH515150 | | | 1 | | | | | 1 |
| Hap_2 | MH515151 | 5 | | 46 | 1 | 18 | 18 | 28 | 116 |
| Hap_3 | MH515153 | | | | | 1 | | | 1 |
| Hap_4 | MH515154 | | | | | 1 | | | 1 |
| Hap_5 | MH515155 | | | | | 1 | | | 1 |
| Hap_6 | MH515156 | | | | | 7 | | | 7 |
| Hap_7 | MH515157 | | | | | 1 | | | 1 |
| Hap_8 | MH515158 | | | | | 1 | | | 1 |
| Hap_9 | MH515159 | | | | | 3 | | | 3 |
| Hap_10 | MH515160 | | | | | 1 | | | 1 |
| Hap_11 | MH515161 | | | | | 1 | | | 1 |
| Hap_12 | MH515162 | | | | | 1 | | | 1 |
| Hap_13 | MH515163 | | | | | 1 | | | 1 |
| Hap_14 | MH515164 | | | | | 1 | | | 1 |
| Hap_15 | MH515165 | | | | | 1 | | | 1 |
| Hap_16 | MH515166 | | | | | 1 | | | 1 |
| Hap_17 | MH515167 | | | | | 1 | | | 1 |
| Hap_18 | MH515168 | | | | | 2 | | | 2 |
| Hap_19 | MH515169 | | | | | 1 | | | 1 |
| Hap_20 | MH515170 | | | | | 2 | | | 2 |
| Hap_21 | MH515171 | | | | | 1 | | | 1 |
| Hap_22 | MH515173 | 9 | | | 32 | | | | 41 |
| Hap_23 | MH515174 | 3 | | | 14 | | | | 17 |
| Hap_24 | MH515176 | | | | | | 29 | 19 | 48 |
| Hap_25 | MH515177 | | | | | | | 1 | 1 |
| Hap_26 | MH515178 | | 48 | | | | | | 48 |
| Hap_27 | MH515179 | | | | | | 1 | | 1 |
| Hap_28 | MH515180 | 15 | | | | | | | 15 |
| Hap_29 | MH515181 | 3 | | | | | | | 3 |
| Hap_30 | MH515183 | 8 | | | | | | | 8 |
| Hap_31 | MH515184 | 3 | | | | | | | 3 |
| Hap_32 | MH515185 | 1 | | | | | | | 1 |

the haplotype was obviously divided into two branches (the AR population was separated from other six populations). In addition, the MS and TW populations were clustered and then clustered with the IL population; GIFT and MY populations were clustered and then clustered with the NL population.

## Genetic differentiation among tilapia populations

The results of the analysis of molecular variance (AMOVA) were shown in Table 4. Based on the results of the genetic differentiation analysis, the variance percentage of genetic variation among populations in total variance was 63.57%, and a high degree of

**Table 2  Genetic diversity parameters of mtDNA D-loop sequence of seven tilapia populations.**

|  | NL | AR | MS | GIFT | TW | MY | IL | Sum |
|---|---|---|---|---|---|---|---|---|
| S | 116 | 0 | 6 | 70 | 101 | 71 | 74 | 146 |
| h | 8 | 1 | 2 | 3 | 20 | 3 | 3 | 32 |
| Hd | 0.826 | 0 | 0.043 | 0.457 | 0.834 | 0.504 | 0.513 | 0.817 |
| $\pi$ | 0.060 | 0 | 0.0003 | 0.004 | 0.024 | 0.039 | 0.040 | 0.054 |
| k | 47.32 | 0 | 0.255 | 3.380 | 20.50 | 33.55 | 34.38 | 45.36 |
| Tajima's D | 2.374[*] | 0 | −2.094[*] | −2.784[**] | −0.464 | 3.874[**] | 3.754[**] | 2.523 |

Notes.

S, polymorphic sites; h, haplotypes; Hd, haplotype diversity; $\pi$, nucleotide diversity; k, average number of nucleotide differences.

[*]means significant ($P < 0.05$).

[**]means extremely significant ($P < 0.01$).

**Table 3  Pairwise K2P genetic distances (below diagonal) and fixation indexes ($F_{ST}$, above diagonal) among seven tilapia populations using D-loop.**

|  | NL | AR | MS | GIFT | TW | MY | IL |
|---|---|---|---|---|---|---|---|
| NL | – | 0.612[**] | 0.639[**] | 0.333[**] | 0.463[**] | 0.181[**] | 0.263[**] |
| AR | 0.070 | – | 0.999[**] | 0.978[**] | 0.876[**] | 0.800[**] | 0.794[**] |
| MS | 0.076 | 0.093 | – | 0.971[**] | 0.128[**] | 0.612[**] | 0.379[**] |
| GIFT | 0.045 | 0.101 | 0.078 | – | 0.798[**] | 0.395[**] | 0.571[**] |
| TW | 0.073 | 0.095 | 0.014 | 0.070 | – | 0.369[**] | 0.133[**] |
| MY | 0.057 | 0.097 | 0.050 | 0.034 | 0.049 | – | 0.079 |
| IL | 0.064 | 0.095 | 0.032 | 0.050 | 0.036 | 0.042 | – |

Notes.

[**]means extremely significant ($P < 0.01$).

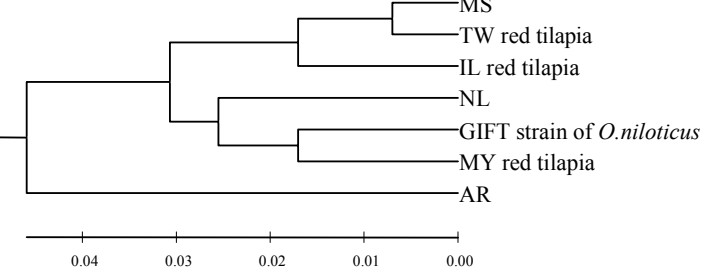

**Figure 1  The UPGMA tree based on the D-loop sequences of seven tilapia populations.** The numbers above the scale line indicate the K2P genetic distances among populations.

inter-population differentiation was observed among populations ($Fst = 0.636$, $P < 0.01$). The seven populations were divided into two groups consisting of wild-type or breeding populations, and red tilapia. The genetic differentiation index among populations within groups accounted for 0.633 ($P < 0.01$). The pairwise $F$-statistics values ($Fst$) of the seven tilapia populations (Table 3, upper right corner) showed that the pairwise genetic differentiations among the populations were very significant ($P < 0.01$), except among the MY and IL populations ($P > 0.05$).

**Table 4** Analysis of molecular variance (AMOVA) of seven tilapia populations based on D-loop sequences.

| Source of variation | d.f. | Sum of squares | Variance components | Percentage of variation | Fixation Index |
|---|---|---|---|---|---|
| No group | | | | | |
| Among populations | 6 | 5848.14 | 20.31 | 63.57 | 0.636[**] |
| Within populations | 325 | 3781.13 | 11.63 | 36.43 | |
| Total | 331 | 9629.27 | 31.94 | | |
| Two groups (Red tilapia and others) | | | | | |
| Between groups | 1 | 900.47 | −0.5567 | −1.760 | 0.639[**] |
| Among populations within groups | 5 | 4947.67 | 20.62 | 65.06 | 0.633[**] |
| Within populations | 325 | 3781.13 | 11.63 | 36.70 | −0.018 |
| Total | 331 | 9629.27 | 31.70 | | |

**Notes.**
d.f., degrees of freedom.
[**] means extremely significant ($P < 0.01$).

## Haplotype network of D-loop sequences in tilapia populations

The median-joining (MJ) network diagram of D-loop haplotypes was described in Fig. 2. The MJ network presented a star-like profile, which was linked to many haplotypes from different regions, and the shared haplotypes and dominant haplotypes were clearly defined. Obviously, three shared haplotypes were composed of two populations (GIFT and NL, MY and IL), one shared haplotype was composed of six populations (except AR) and a dominant haplotype was composed of an AR population.

## Maternal ancestry of red tilapia strains

According to the genetic distances among the tilapia populations (Table 3, below diagonal), it was calculated that TW, IL red tilapia, and MS populations had the closest genetic distance, which were 0.014 and 0.032, respectively. The genetic distance between MY red tilapia and the GIFT population was the closest (0.034). Moreover, the UPGMA tree clearly divided the tested samples of red tilapia into two independent branches. TW and IL were clustered into one branch, and then clustered with MS, while MY divided into another branch and clustered with GIFT and NL.

## DISCUSSION

### Genetic diversity and population dynamics

In the present study, the content of A+T (64.3%) in tilapia D-loop sequences was higher than the content of G+C (35.7%), which was in line with the distribution characteristics of the base content in the D-loop (control region) of many fishes (*Broughton, Milam & Roe, 2001*). In D-loop sequences of seven tilapia populations, 146 polymorphic (segregating) sites (S) and 32 haplotypes were detected, suggested that D-loop sequences could be an effective marker for genetic diversity analysis for tilapia populations. Overall, the tilapia populations showed high haplotype diversity and nucleotide diversity in this study, indicating the populations that contain an abundance of genetic resources for further use in breeding or practice. Specifically, NL had the higher genetic diversity (Hd >0.5, $\pi$ >0.005),

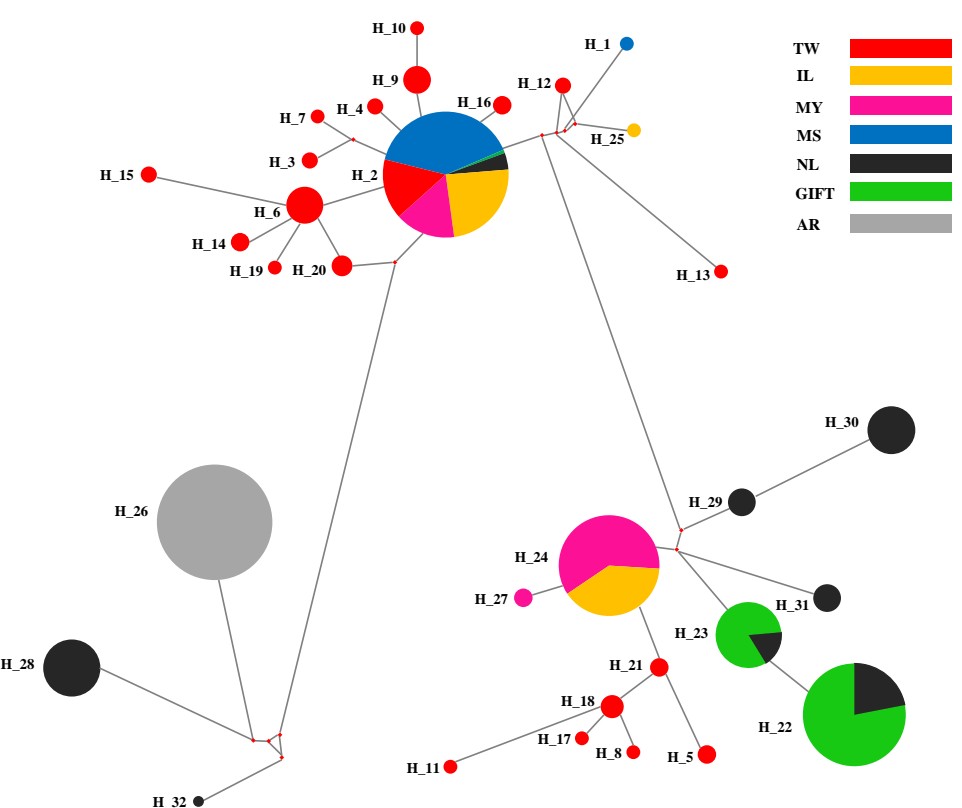

**Figure 2 The haplotypes network of the D-loop sequences for seven tilapia populations.** Different populations indicated with different color; the pie sizes mean the distribution frequencies of haplotypes in populations.

which was consistent with other research using microsatellite markers (*Romana-eguia et al., 2004*; *Yang et al., 2011*), and isozymes (*Zhao et al., 1997*). It was speculated that the original NL population introduced in China was larger and it had a potential for further selective breeding. The genetic diversity of the red tilapia populations (TW, IL, and MY) was higher than the other tilapia populations, probably due to its genetic background of cross-breeding. However, the genetic diversity of the AR population was the lowest ($Hd = 0$, $\pi = 0$), which was found to be similar to the results of the previous reports assessed by RAPD (*Xia et al., 1999*), microsatellite probes (*Wang, Xia & Wu, 2000*), and mtDNA restriction enzyme analysis (*Cao & Xia, 1997*). This may be due to the small population size introduced in China as well as mass breeding over generations, which might result in a decline in the genetic polymorphism of the population. The genetic purity of the AR population was adverse to further selective breeding and it is necessary to introduce the AR population again in order to improve its genetic diversity and reduce its depression as a result of inbreeding. In contrast, the purity of this population also could be used for inhybridization with other populations or strains. The GIFT and MS populations were also detected with low diversity parameters ($Hd < 0.5$, $\pi < 0.005$), indicating that its population may have recently experienced a bottleneck or founder effect produced by

minority populations (*Grant & Bowen, 1998*). The high purity of the AR, GIFT, and MS populations has great significance that the excellent economic traits obtained through long-term multi-generation breeding could insure the stability inherited through the genetic selection process. Therefore, AR, GIFT, and MS were often used as parents to breed red tilapia and stabilize the excellent traits in the red tilapia populations. The Tajima's D value for some red tilapia (TW) and wild-type or breeding populations (MS, GIFT) were negative, which is probably due to the population expansion caused by larger scale breeding after smaller breeder population produced a family selection in the hatchery.

## Genetic relationships of tilapia populations

The red tilapia stocks were reported with different genetic sources that originated from *O. mossambicus, O. aureus,* or *O. niloticus* (*Wohlfarth et al., 1990*; *Sandeep et al., 2012*). In this study, four relative tilapia populations were used for exploring the maternal ancestors for three strains of red tilapia. The MJ network of D-loop sequences for haplotypes in the tilapia populations, which were divided into three different major regions and red tilapia populations, presented in two regions. Four dominant haplotypes were shared by the seven populations, where Hap_24 was a shared by MY and IL populations, Hap_22 and Hap_23 were shared by GIFT and NL populations, and Hap_2 was shared by six populations except for the AR population. It was suggested that six populations (except AR) may originate from similar maternal ancestors.

The analysis of molecular variance (AMOVA) by grouping and non-grouping showed that the main genetic variation was derived from inter-population, which is similar to the results of *Habib et al. (2011)*, the low variance within the population and high variance among populations was reported among *Channa* fishes. The fixation index ($F$st) was commonly used to examine the genetic variation of populations and the contribution of this variation to genetic differentiation (*Holsinger & Bruce, 2009*). Significant $F$st values ($F$st $>0.25$, $P < 0.01$) were found in this study, which demonstrated that a higher level of genetic differentiation exists among tilapia populations except for red tilapia. The results indicated that red tilapia populations may have evolved independently after separating from a common ancestor, but that those three strains of red tilapia were closed.

## Analysis of maternal ancestry of red tilapia

While the maternal ancestors of the existing three strains of red tilapia are not well documented, their derivation is generally attributed to the crossbreeding of the mutant reddish-orange *Oreochromis mossambicus* with other species including *O. aureus, O. niloticus,* and *O. hornorum* (*Wohlfarth et al., 1990*; *Sandeep et al., 2012*). Mitochondrial DNA (mtDNA) has the characteristics of maternal inheritance, simple structure, and rapid evolution. Therefore, the phylogenetic tree constructed by mtDNA can directly reflect the origin of the maternal ancestry (*Cann, Stoneking & Wilson, 1994*).

Based on the K2P genetic distances among seven tilapia populations, two branches were constructed in the UPGMA dendrogram and it was speculated that these populations might be derived from two different primary maternal ancestors, which was consistent with the results of the MJ network. Specifically, three strains of red tilapia might derive

from different maternal origin, MS and GIFT (or NL) populations, respectively. The TW and IL populations were closely related to each other and were clustered with MS, which was confirmed that two strains of red tilapia were produced from local crossbreeding of the rare, mutant-colored (reddish-orange) female *O. mossambicus* (*Wohlfarth et al., 1990*). The GIFT strain was selected from four *O. niloticus* strains imported directly from Africa and four strains widely cultivated in Asia (*Eknath et al., 1993*). The K2P genetic distance between MY red tilapia and the GIFT population was relatively small ($D = 0.034$), speculating that MY population was probably bred with the GIFT population or that MY and GIFT populations might come from the similar, artificially selected NL population. In addition, the degree of genetic differentiation between red tilapia (IL, TW, and MY) and the breeding source populations (MS, GIFT or NL) were relatively small, demonstrating that a close genetic relationship was maintained between the breeding varieties and breeding source populations; this highly homology was related to the characteristics of maternal inheritance and non-recombination of mtDNA (*Mabuchi, Senou & Nishida, 2010*).

## CONCLUSIONS

In this study, we used the D-loop sequences to estimate the genetic structures of seven tilapia populations mainly cultured in China. Furthermore, we analyzed the maternal ancestry of three strains of red tilapia, which provides more basic data for the reasonable protection and further utilization of tilapia populations in the future. In brief, the IL and TW red tilapia strains may derive from the *O. mossambicus* population, whereas the MY red tilapia was probably derived from GIFT or *O. niloticus*.

### Funding

This work was supported by the Central Public-interest Scientific Institution Basal Research Fund from Chinese Academy of Fishery Sciences (2017HY-XKQ0203) and the Jiangsu Natural Science Foundation for Young Scholar (BK20160203). The funders had no role in study design, data collection and analysis, decision to publish, or preparation of the manuscript.

### Grant Disclosures

The following grant information was disclosed by the authors:
Central Public-interest Scientific Institution Basal Research Fund from Chinese Academy of Fishery Sciences: 2017HY-XKQ0203.
Jiangsu Natural Science Foundation for Young Scholar: BK20160203.

### Competing Interests

The authors declare there are no competing interests.

### Author Contributions

- Bingjie Jiang performed the experiments, analyzed the data, prepared figures and/or tables and authored the paper.

- Jianjun Fu, Zaijie Dong conceived and designed the experiments, contributed reagents/materials/analysis tools, reviewed drafts of the paper, approved the final draft.
- Min Fang performed the experiments, collect samples.
- Wenbin Zhu and Lanmei Wang contributed reagents/materials/analysis tools.

**Animal Ethics**

The following information was supplied relating to ethical approvals (i.e., approving body and any reference numbers):

All sampling scheme and experimental protocols were approved by the Bioethical Committee of Freshwater Fisheries Research Center (FFRC) of Chinese Academy of Fishery Sciences (CAFS) (BC2013863, 9/2013). The methods of all animals were handled and experimental procedures carried out in accordance with the guidelines for the care and use of animals for scientific purposes set by Ministry of Science and Technology, Beijing China (No. 398, 2006).

**DNA Deposition**

The following information was supplied regarding the deposition of DNA sequences:

Data is available at GenBank, accession numbers: MH515150–MH515185 (except for MH515152, MH515172, MH515175, and MH515182).

**Data Availability**

The new DNA sequences are available in the Supplemental File.

**Supplemental Information**

Supplemental information for this article can be found online at http://dx.doi.org/10.7717/peerj.7007#supplemental-information.

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
