# Peer review of "Maternal ancestry analyses of red tilapia strains based on D-loop sequences of seven tilapia populations"

_PeerJ, doi:10.7717/peerj.7007_

## Round 0.1 · original submission · Minor Revisions

· Academic Editor

Minor Revisions

Please provide a response indicating how you addressed each of the reviewer's comments. Note that reviewer 1 provided an annotated copy of the manuscript with comments and suggestions.

·

Basic reporting

No comment

Experimental design

No comment

Validity of the findings

No comment

Additional comments

The authors should carefully go through the manuscript again and ensure that due attention is given to the point raised as indicated in the comment boxes attached to the manuscript.

Reviewer 2 ·

Basic reporting

no comment

Experimental design

no comment

Validity of the findings

no comment

Additional comments

The manuscript title with ‘Maternal ancestry of red tilapias based on D-loop sequences analysis of seven tilapia populations’ written by Jiang et al analyzed the genetic diversity and variants of seven tilapia populations, and further speculated the maternal ancestries of three red tilapia varieties, based on the mtDNA D-loop sequences. The paper finely descripted the experiment design, and also provided a relatively complete genetic profile for the tilapia in China, especially demonstrated the new genetic information for the maternal ancestries of several red tilapia strains. In brief, the research is valuable for understanding the genetic profile of tilapia in China, especial for genetic origins of red tilapias, which could be the support data for the genetic protection and further utilization of tilapia resources in China. However, there are some mistakes and unclear descriptions presented in this paper.
In my personal opinion, the present paper could be accepted only after minor revision.
The specific advises or suggestions are written as follow:
(1) Line 29: the description of tilapia resource of China, which might not closely with the present study. Which would suggest to replace with ‘many tilapia species or varieties have been widely ….’.
(2) Line 36: the scientific name of red tilapia had been written with mistake, which should be corrected with ‘Oreochromis spp.’.
(3) Line 63: there are words presented not clear, that is ‘tilapia variants that sport and attractive red coloration.’
(4) Line 80: the three strains are not clear, which are first time presented in the introduction. It is suggested to be rewritten with ‘three strains widely cultured in China (Yang et al., 2016)’.
(5) Line 98: the number of ‘48’ in the head of sentence, should be corrected using the letter form, i.e. forty-eight.
(6) Lines 125 & 128: the written form of software names should be checked carefully, ‘DNAsp5.1’ and ‘Mega 5.05’, should be rewritten with ‘DnaSP 5.1 software’ and ‘MEGA 5.05’.
(7) Lines 128-131: the sentence which suggested to be rewritten with ‘MEGA 5.05 software (--) was utilized to calculate the Kimura 2-parameter (K2P) distances among populations, and to construct the Neighbor-joining (NJ) dendrogram set with 1000 replications of bootstrapping.’
(8) Line134: the software used to construct the haplotypes network, which should be provided in the paper.
(9) Lines 140-142: the description for nucleotide composition of D-loop sequence is incorrect. The D-loop sequence is one kind of double-strand DNA sequence, the composition of A and C should equal to T and G, respectively.
(10) Lines 175-180: the word ‘inter-population’ which suggested to be replaced with ‘among populations’.
(11) Line 184: the subtitle of this paragraph, which suggested corrected with ‘haplotypes network of D-loop sequences in tilapia populations’.
(12) Line 219: the word ‘generation’ should be replaced with ‘generations’.
(13) Line 248 & 251: corrected the genus name and significant probability symbol with italic font, i.e. Channa, and P, respectively.
(14) References related advices, the form of reference cited in manuscript and the references list should be carefully checked according to the journal. such as following:

---

## Round 0.2 · Minor Revisions

· Academic Editor

Minor Revisions

Thank you for responding to the reviewers' comments and for providing a revised version of your manuscript.

Although the scientific content is now Acceptable, a final check by staff and the Section Editors has revealed that the English language needs further improvement before the article can be accepted. Therefore we are returning it to you to perform an additional language edit.

---

## Round 0.3 · accepted · Accept

· Academic Editor

Accept

Thank you for attending to the outstanding language issues which have now been addressed.

#